# Oncoplastic Breast Surgery versus Conservative Mastectomy in the Management of Large Ductal Carcinoma In Situ (DCIS): Surgical, Oncological, and Patient-Reported Outcomes

**DOI:** 10.3390/cancers14225624

**Published:** 2022-11-16

**Authors:** Elena Jane Mason, Alba Di Leone, Antonio Franco, Sabatino D’Archi, Chiara Rianna, Alejandro Martin Sanchez, Federica Murando, Cristina Accetta, Lorenzo Scardina, Daniela Andreina Terribile, Riccardo Masetti, Gianluca Franceschini

**Affiliations:** 1Fondazione Policlinico Universitario Agostino Gemelli IRCCS, Multidisciplinary Breast Center, Department of Woman and Child Health and Public Health, 00168 Rome, Italy; 2Facoltà di Medicina e Chirurgia, Università Cattolica del Sacro Cuore, 00168 Rome, Italy; 3Facoltà di Medicina e Chirurgia, Istituto di Semeiotica Chirurgica, Università Cattolica del Sacro Cuore, 00168 Rome, Italy

**Keywords:** breast cancer, large DCIS, in situ, breast-conserving therapy, oncoplastic surgery, mastectomy, patient-reported outcomes, BREAST-Q, immediate breast reconstruction

## Abstract

**Simple Summary:**

The safety and efficacy of level II oncoplastic surgery (OPS2) is well-established in the treatment of invasive breast cancer, and most research suggests a benefit, compared to mastectomy, in terms of patient satisfaction and final breast cosmesis. However, very few studies address its use in the treatment of large DCIS, a condition with a unique clinical presentation, which often represents a surgical challenge, given the intraoperative difficulty in assessing disease margins. Our aim was to analyze the surgical, oncological, and patient-reported outcomes in patients undergoing oncoplastic level II breast-conserving surgery for large DCIS, to assess its safety, efficacy and final cosmetic outcome. This is the first study on large DCIS to compare OPS2 with conservative mastectomy, the other option a surgeon can offer to these patients. It is also the first to include patient-reported outcomes, and does so by means of a standardized instrument such as BREAST-Q©.

**Abstract:**

Oncoplastic level II breast-conserving surgery (OPS2) allows for wider excisions than standard breast-conserving surgery, but the literature on this technique in the treatment of DCIS is scarce. This study compares OPS2 to conservative mastectomy (CM) in patients undergoing surgery for large DCIS. The clinical, radiological, surgical, and post-operative data of 147 patients who underwent either CM or OPS2 for large DCIS between 2007 and 2021 were retrospectively reviewed. The surgical, oncological, and patient-reported outcomes (PRO) were analyzed and compared between the two groups. The surgical outcomes were similar, in terms of margin involvement (*p* = 0.211), complication rate (*p* = 0.827), and re-excision rate (*p* = 1). The rate of additional surgery for cosmetic optimization was significantly lower in the OPS2 group: only 1 (1.8%) patient required surgical adjustments versus 24 (26.4%) patients in the CM group (*p* < 0.001). The mean hospital stay was lower in the OPS2 group (*p* < 0.001). The oncological outcomes did not differ between the two groups (*p* = 0.662). The PRO analysis showed better outcomes in the OPS2 group, which achieved statistical significance in the *sexual well-being* module (*p* = 0.015). Skin sensitivity loss was also significantly lower in the OPS2 group (*p* < 0.001). When feasible, OPS2 should be considered in the treatment of large DCIS, as it is safe and shows high levels of patient satisfaction.

## 1. Introduction

Breast cancer is a widespread disease affecting 1 in 8 women worldwide during their lifetime [1]. The wider utilization of screening mammography over the last 30 years has led to a growing detection of early breast cancers, and today, approximately 20% of new breast cancer diagnoses refer to ductal carcinoma in situ (DCIS) [2]. 

Most DCIS are identified through mammograms, as nearly 90% show the presence of microcalcifications, with or without a concurrent breast mass [3]. The more extensive the lesion, the higher the probability of finding an invasive component at the final histology [4]. As a result, although DCIS is common per se, the finding of a pure, extensive in situ disease is not so frequent. 

During the last 20 years, both mastectomies and breast-conserving procedures have become less invasive, in order to better comply with psychological and cosmetic requirements. The term “conservative mastectomy” refers to a tissue-sparing technique that achieves complete glandular excision, with the preservation of the skin envelope and, when possible, of the nipple–areolar complex (NAC), allowing for an immediate and more cosmetic breast reconstruction [5]. Breast-conserving surgery (BCS) has also expanded thanks to the use of level II oncoplastic techniques, which allow for resections of large lesions involving up to 50% of the total breast volume, without compromising the final breast cosmesis [6]. These techniques have reduced both the mastectomy and re-excision rates; however, their use in the management of DCIS was reported in only a few studies, with a small series of patients [7,8,9,10,11]. 

## 2. Materials and Methods

This study was conducted in accordance with the ethical standards of our university, as laid down in the Declaration of Helsinki, and within a protocol approved by the central ethics committee (protocol registration ID: 4668). We retrospectively reviewed all women with a preoperative diagnosis of DCIS treated at our Institution between 1 January 2007 and 31 December 2021. Patients subjected to either conservative mastectomy (CM) or level II oncoplastic breast-conserving surgery (OPS2) with a final diagnosis of pure or microinvasive DCIS were included in the study. Exclusion criteria were history of metachronous or synchronous contralateral invasive breast cancer, level I oncoplastic BCS, radical mastectomy without reconstruction, and evidence of invasive breast cancer at final histology report. 

### 2.1. Clinical Management

Preoperative workup included breast ultrasound, mammograms, and clinical evaluation for all patients. Breast MRI was performed in patients with dense breasts to better assess disease extension. Histologic confirmation of DCIS was acquired preoperatively in all cases with either ultrasound guided core-needle biopsy, vacuum-assisted stereotactic or MRI-guided breast biopsy, or excisional biopsy. Indication to level II oncoplastic surgery was given when the total resection volume was estimated between 20% and 50% of the glandular volume, while CM was performed if the estimated excision volume was >50%, if the surgeon expected a poor aesthetic outcome despite the use of OPS2 (such as in small breasts), or in case of patient preference. All surgical indications were discussed during a preoperative multidisciplinary meeting, which included breast surgeons, plastic surgeons, pathologists, and radiologists. 

In all patients undergoing OPS2, tumor localization was performed with ultrasound or stereotactic-guided skin projection tattoo [12]. 

In patients with preoperative evidence of microcalcifications, specimen x-ray was performed routinely in the OPS2 group and in case of superficial or peripheral microcalcifications in the CM group. Additional margin shavings were acquired in both groups when intraoperative examination was suggestive of involved margins. 

In the CM group, immediate final breast reconstruction was performed whenever possible, with either implant positioning (prepectoral or submuscular) or flap reconstruction. A tissue expander was positioned only when the final aesthetic result of an immediate definitive reconstruction was considered inadequate. In the OPS2 group, plastic surgical strategy included T and J bilateral mammoplasties and round block, batwing, and racquet unilateral or bilateral procedures. Postoperative complications were defined as complications arising within 30 days after surgery, and severity was ranked using the Clavien–Dindo classification [13].

Specimen margins were defined as involved in cases of DCIS presence in the cut edge of the specimen (ink on tumor) and close in cases of tumor cells within 2 mm of the edge. Indication to adjuvant treatments was discussed in all cases at a multidisciplinary meeting after the acquisition of final histology and based on NCCN guidelines [14]. 

Patient-reported outcomes were collected using 4 modules of the Memorial Sloan Kettering BREAST-Q© questionnaire (Italian version 2.0), after at least 6 months from surgery or the end of radiotherapy. As there is no specific BREAST-Q© module to explore residual skin sensitivity after breast surgery, we developed a brief questionnaire consisting of 4 questions investigating this aspect. Patients were asked to provide a score between 1 and 10 on local sensitivity loss, and its impact on sexual and everyday life. Follow-up was performed every 6 months for 5 years, then annually.

### 2.2. Statistical Analysis

Statistical analysis was performed using SPSS© software for Windows, version 24.0. Continuous variables were described by mean (median; interquartile range) and categorical variables, such as number and percentage. Disease-free survival plots were drawn using the Kaplan–Meier method, and differences were determined with the log rank test. Data distribution between the two groups was compared using Chi-square test for categorical variables and Fisher’s exact test for small-sized samples. Continuous variables were tested for normality using the Shapiro–Wilk test, and Mann–Whitney U test was selected to compare distribution among the two groups. Statistical significance was achieved with *p* values < 0.05. All *p* values were two-tailed. 

## 3. Results

### 3.1. Patients and Tumor Characteristics

The initial analysis identified 943 patients with a breast biopsy showing DCIS. Of these, 81 patients had a history of metachronous or contralateral synchronous invasive breast cancer and were excluded from the study. As this study considers only women treated with either conservative mastectomy (CM) or level II oncoplastic breast-conserving surgery (OPS2), we excluded 611 patients that underwent other kinds of conservative or radical surgical procedures. In 104 patients, final histology revealed DCIS with concurrent invasive cancer; therefore, these cases were also excluded from the study. At a final analysis, this research included 147 patients with large pure DCIS: 91 patients that received a conservative mastectomy (CM group—61.9%) and 56 patients that received a level II oncoplastic breast-conserving surgery (OPS2 group—38.1%) (Figure 1). In the OPS2 group, the majority of patients were treated with a bilateral mammoplasty. In the CM group, the majority of patients underwent a nipple-sparing mastectomy with immediate prosthetic breast reconstruction (IPBR). Surgical treatment specifics are summarized in Table 1.

Patient preoperative characteristics did not differ significantly between the two groups. Overall mean lesion extension was 48.9 mm (43 mm; 25–70 mm) at preoperative imaging, with no significant difference between the two groups (*p* = 0.207), and 43.4 mm (38; 25–55) at final histology, with patients in the CM group showing slightly larger lesions (*p* = 0.037). Tumor characteristics were comparable between the two groups of study. Patient and tumor characteristics are summarized in Table 2 and Table 3.

### 3.2. Surgical Outcomes

Final margin assessment showed clear margins in 65 (71.4%) patients in the CM group and 45 (80.4%) patients in the OPS2 group. Margins were close (<2 mm) in 18 (19.8%) mastectomy specimens and 5 (8.9%) OPS2 specimens and involved (ink on tumor) in 8 (8.8%) mastectomy and 6 (10.7%) OPS2 specimens.

Hospital stay was shorter in the OPS2 group (*p* < 0.001). Out of 25 total perioperative complications, of which 16 (17.6%) were in the CM group and 9 (16.1%) were in the OPS2 group, 20 were classified as mild, with a Clavien–Dindo score of 1, and two were classified as moderate (Clavien–Dindo 2). Three patients developed a Clavien–Dindo score 3 complication requiring surgical revision: one patient from each group for a post-operative hematoma and another patient in the CM group for an implant infection. 

Overall, no statistical differences were found in the final margin assessment (*p* = 0.211) and complication rates (*p* = 0.827) between the two groups. 

Patients treated conservatively with close or focally involved margins were discussed at a multidisciplinary meeting to determine the need for further surgery: in 8 out of 11 patients margin involvement was minimal, and patients were counselled for adjuvant whole breast irradiation. In the remaining three patients, re-excision was suggested; however, two patients refused to undergo a new surgery. Overall, one patient in each group underwent re-excision for margin involvement: one woman in the CM group was subjected to a NAC excision after discovery at final histology of an involved retroareolar tissue margin, and one patient in the OPS2 group underwent a re-quadrantectomy (*p* = 1.000). No patient underwent completion mastectomy following primary OPS2. 

Most additional surgeries were performed for cosmetic adjustments, a feature significantly more predominant in the CM group, where 24 (26.4%) patients underwent at least one new surgery, compared to only 1 (1.8%) patient in the OPS2 group (*p* < 0.001). Overall, the second surgery rate was significantly higher in the CM group, with 30 patients (33%) undergoing at least one new operation, compared to only 4 patients (7.1%) in the OPS2 group (*p* < 0.001).

### 3.3. Oncological Outcomes

The mean follow-up was 60.5 (55.7; 23.6–89.6) months in the OPS2 group and 34.5 (32; 10–50) months in the CM group. No death or systemic recurrence was observed during the follow-up period. Three women in the CM group (3.3%) and one in the OPS2 group (1.8%) developed a local recurrence (LR = 0.407) (Figure 2).

In the CM group, two recurrences occurred on the NAC in the form of Paget’s disease and were successfully treated with nipple excision. Another patient developed a recurrence on the skin-flap of the upper-outer quadrant that was re-excised and showed DCIS at final histology. Out of the three patients in the CM group who developed a local recurrence, in two cases, the primary mastectomy had resulted in clear margins, while in one mastectomy specimen, the margins were close. 

In the OPS2 group, the only observed recurrence occurred 24 months after BCS with bilateral mammoplasty. Primary surgery histology showed a high-grade pure DCIS, HER2 positive, and hormone receptor negative. Specimen margins were focally involved. The patient was counselled for completion mastectomy, but refused both further surgery and adjuvant radiotherapy. Recurrence histology showed an invasive ductal carcinoma. 

Surgical and oncological outcomes are summarized in Table 4.

### 3.4. Aesthetic and Functional Outcomes

Out of 147 enrolled patients, 124 women voluntarily completed the PRO questionnaires. Patients were administered BREAST-Q© questionnaire modules on psychosocial well-being, physical well-being, sexual well-being, and satisfaction with breasts. The subsequent analysis of these data showed high levels of satisfaction in both groups; however, women from the OPS2 group were more likely to report better scores. A statistically significant difference between the scores reported by the two groups of study was noted in the sexual well-being module (*p* = 0.015). 

Residual skin sensitivity was investigated with an additional brief questionnaire developed by our team. Scores were compared between the two groups, and a statistical significance for better outcomes in the OPS2 group was achieved for all questions. The results are summarized in Table 5.

## 4. Discussion

DCIS is a heterogeneous condition with a wide range of clinical presentations. Recurrence rates range between 2% and 16%, depending on treatment [15,16]. Most recurrences occur in the same breast, and approximately half of them are invasive [17]. Surgery and radiotherapy, therefore, play a key role in the management of DCIS and must be planned accurately to guarantee an optimal local radicality and prevent invasive recurrences that can lead to the systemic spread of disease. 

The great majority of DCIS lesions can be managed conservatively, and the efficacy of BCS with adjuvant radiotherapy is well-established [18]. However, adequate excision of large DCIS using traditional BCS may compromise the final cosmetic result. Moreover, large DCIS is often a disease with distinct pathological features and high-risk lesions showing either high-grade comedonecrosis or microinvasion are more common than in smaller DCIS [19,20,21]. This was also evident in our population of study, where 78.6% of patients presented at least one of these characteristics at final histology (Table 3). Axillary involvement is also of note, as in our overall population, 10 patients (6.8%) had evidence of nodal disease at final histology, of which only 2 with microinvasive DCIS. Although the total number of patients was too small to provide any statistically significant evidence, the rate was much higher than that reported in a recent, large cohort study by James et al. on more than 15,000 patients with pure DCIS undergoing BCS, of which only 0.9% had evidence of nodal involvement at final histology [22]. 

The best surgical strategy for large DCIS, in order to guarantee a safe oncological outcome with minimum physical and psychosocial impact, is, therefore, still a cause of debate. To our knowledge, this is the first study comparing conservative mastectomy with oncoplastic level II breast-conserving surgery, the only two options that a breast surgeon can offer a woman with extensive DCIS to provide both oncological radicality and an optimal cosmetic outcome. It is also the first study to take into account patient-reported outcomes, such as cosmetic results and impact on everyday life, which were included in our analysis by means of a standardized tool, as is the BREAST-Q© [23], and expressed a holistic evaluation of the women’s well-being.

### 4.1. Oncological Radicality

Margin involvement has long been the object of study in both mastectomy and lumpectomy patients: more than 70% of DCIS are multifocal, with single lesions often smaller than 5 mm [24], a feature that exposes to the risk of a small margin falling within the healthy tissue between two lesions [25]. Current NCCN guidelines recommend the pursuit of a tumor-to-margin distance of at least 2 mm in patients undergoing BCS and adjuvant radiotherapy, although patients with evidence of minimal margin involvement can avoid re-excision in selected cases and after accurate clinical judgement [14]. 

Only a few studies have analyzed margin assessment in DCIS patients treated with OPS2. Two studies on a small series of patients reported positive margin rates of 13% and 32% and only included margins smaller than 1 mm, suggesting that a higher percentage of women could have close margins between 1 mm and 2 mm [7,8]. More recently, a study by the European Institute of Oncology reported close or focally involved margins in 12 out of 44 patients treated with OPS2 and adjuvant radiotherapy, even though the authors excluded from their series women who underwent OPS2, but were subsequently treated with re-excision or completion mastectomy. In their analysis, margin status was not associated with an increased risk of local recurrence [10]. A 2019 study by Van la Parra and colleagues examined a large case series of 68 patients and found a positive margin rate of 14.7%, significantly related to a tumor size larger than 5 cm [9]. Finally, a recent study by Crown et al. reported a 16% close margins rate, although none of the 50 patients presented focally involved margins, and no recurrences were noted during a median follow-up of 46 months [11]. 

Margin assessment in DCIS patients treated with mastectomy is also controversial. Current NCCN guidelines recommend considering chest wall irradiation in patients with invasive breast cancer and involved or close mastectomy margins; however, no indication is offered in the case of mastectomy for DCIS. Few single-institution studies with heterogeneous results are available in the literature, and the involved margin rates varied between 14% and 43.8% [26,27,28,29]. A recent metanalysis by Kim et al. studying 2902 women undergoing mastectomy for DCIS found that, although local recurrence in these patients is generally low, margin status is significantly associated with its occurrence. Women with margins closer than 2 mm had a 3.72-fold higher risk of developing a local recurrence, compared to those with negative margins, and patients with margins closer than 1 mm were nearly 3 times more likely to develop a recurrence than patients with close margins wider than 1 mm [30]. Conservative mastectomies are becoming growingly diffuse, and the recent trend towards prepectoral reconstructive techniques, which encourages surgeons to obtain a mastectomy flap thickness wider than 1 cm, suggests that the risk of leaving a residual glandular tissue could be growing accordingly [31]. Therefore, margin assessment in mastectomy specimens cannot be neglected.

Our case series of 56 patients undergoing OPS2 for large DCIS is amongst the largest, and our close or positive margin rates of 19.6% in the OPS2 group and 28.6% in the mastectomy group are in line with the previously reported results. Though recurrences in our population of study were too few to determine a correlation with margin status, our results show that margin involvement rates are similar between the two techniques; therefore, concern for oncological safety should not influence the surgeon’s choice when selecting the best procedure for a patient with large DCIS. 

Two patients with evidence of multicentric disease in their preoperative exams were considered eligible for OPS2, and in both cases, final margin assessment showed negative margins, although in one patient, multicentricity was not confirmed at final histology. On the other hand, out of the three OPS2 patients with evidence of multicentricity at final histology, two had involved margins. In both cases, preoperative exams suggested neither multifocal nor multicentric disease; therefore, OPS2 was conducted upon incorrect preoperative assumptions. Although patients are too few to provide evidence of the efficacy of OPS2 in patients with multicentric disease, the data suggest that these patients could be safely treated with OPS2 in selected cases. Multifocal and multicentric disease invisible at preoperative exams is an unfortunate issue common to all breast-conserving surgery and should not dissuade the surgeon from conserving the breast whenever possible.

### 4.2. Additional Surgeries for Cosmetic Optimization

In our cohort, most additional surgeries were performed for cosmetic adjustments. This occurrence was rare in the OPS2 group, where only one patient returned to the operating room 4 years after the primary surgery for a breast ptosis requiring a new mastopexy. Conversely, patients treated with mastectomy were significantly more at risk of undergoing further surgical treatment, and 24 patients (26.4%) underwent at least one new operation. In twelve mastectomy patients, immediate prosthetic breast reconstruction (IPBR) was not possible; therefore, a tissue expander was positioned, and a delayed prosthetic reconstruction was performed several months after the first surgery. One patient suffered prosthetic infection and underwent several additional surgeries to remove the implant, position a tissue expander, and finally, reconstruct the breast with a tissue flap. In three cases, the final aesthetic result was deemed unsuccessful, and patients were scheduled for a new reconstructive surgery using a musculocutaneous flap. One patient required implant repositioning following accidental displacement. Finally, minor second surgeries in the mastectomy group included aesthetic adjustments, such as lipofilling and wound revisions. 

Additional surgeries have a significant impact both on patient quality of life and healthcare costs. While no investigation was conducted to specifically address these issues, our study suggests that patients treated with OPS2 are less likely to face physical and psychological distress due to a prolonged treatment period with multiple operations, and healthcare expenses may also benefit from the use of these techniques. 

### 4.3. Physical and Psychosocial Impact

Though breast surgery, compared to other surgeries, has a low perioperative impact (short hospital stay, manageable postoperative pain, low morbidity), long-term outcomes can influence significantly the patient’s functional and psychosocial life. Oncoplastic and reconstructive techniques help to contain the personal wound, which could stem from a surgery that often feels like a mutilation, but functional problems, such as chronic chest wall pain or upper limb impairment, and psychological or sexual issues, due to changes in self-image, are not uncommon. 

The Memorial Sloan Kettering BREAST-Q© questionnaire is a validated instrument for quantifying surgical outcomes from the patient’s perspective [23]. We administered the modules on psychosocial well-being, physical well-being, sexual well-being, and satisfaction with breasts. However, many patients reported dissatisfaction with their residual breast skin sensitivity, an issue that is not clearly addressed in the BREAST-Q©. We, therefore, created a brief questionnaire to quantify the sensitivity loss and investigate its impact on the patient’s sexual and everyday life. 

A study by Fosh et al. reported high levels of satisfaction in DCIS patients treated with partial mastectomy and OPS2 [32], and a recent study by Fazeli and colleagues interviewed 316 women with DCIS prior to surgery and found that 80% would have preferred wide local excision over mastectomy as a treatment of choice [33]. Our data suggests that breast-conserving surgery with OPS2 yields lower morbidity than conservative mastectomy, a finding that is in line with that reported by Livingston-Rosanoff and colleagues, who found that DCIS patients treated with mastectomy were more likely to report lower levels of satisfaction than patients treated with BCS [34]. In our cohorts, patients treated with OPS2 reported better outcomes than CM patients in all aspects studied by the BREAST-Q©, particularly concerning their sexual well-being, a feature which is likely due to the influence of residual breast sensitivity on this domain. In fact, perceived residual sensitivity was shown to be significantly higher in the OPS2 group than in the CM group, as was its influence on every aspect of the patients’ lives.

### 4.4. Strenghts and Limitations

To the authors’ knowledge, this is the first study comparing mastectomy with level II oncoplastic surgery in the treatment of large DCIS. The analysis was conducted in a holistic fashion, exploring every aspect of the patient’s wellbeing, from surgical and oncological success to patient-reported satisfaction. To avoid bias, subjective outcomes were collected by means of a standardized tool, such as BREAST-Q©, and further detail has been acquired with a dedicated questionnaire. 

The main limitations of this study reside in its retrospective nature and in the total sample size. However, the total number of patients was larger than most previous reports and consistent with a relatively uncommon disease: in fact, nearly half of the patients who underwent CM or OPS2 for a preoperative diagnosis of extensive DCIS were found to have an invasive component at final histology and excluded from our study. Therefore, our population of study was highly selected and comprised patients showing a disease with unique features. Finally, median follow-up was relatively short in the CM cohort, a feature which could have influenced the low number of local recurrences. However, while this may affect the oncological outcome, surgical and patient-reported outcomes are not significantly determined by the follow-up period.

## 5. Conclusions

The results of our study suggest that breast-conserving surgery with OPS2 is safe and feasible in selected patients with large DCIS, as it provides results comparable to CM, in terms of oncological and surgical efficacy. In these patients, level II oncoplastic surgery can ensure lower patient distress, higher cosmetic satisfaction, and containment of healthcare costs, and therefore, represents a valid alternative to conservative mastectomy.

## Figures and Tables

**Figure 1 cancers-14-05624-f001:**
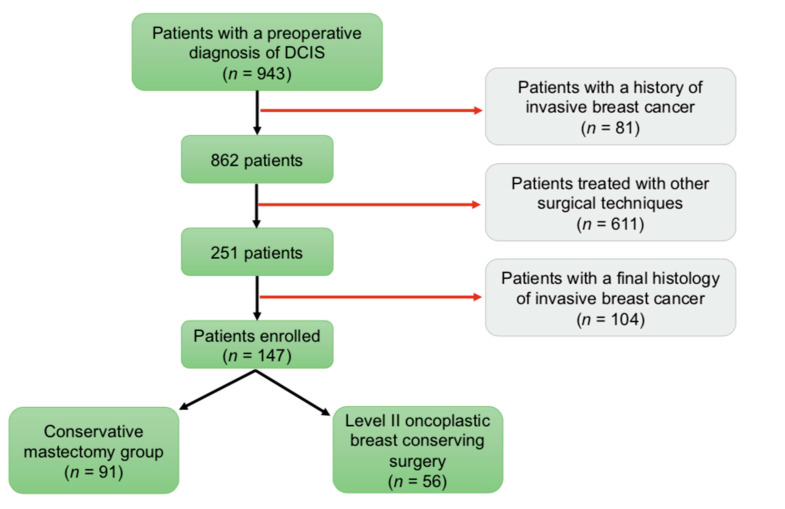
Patient Accrual.

**Figure 2 cancers-14-05624-f002:**
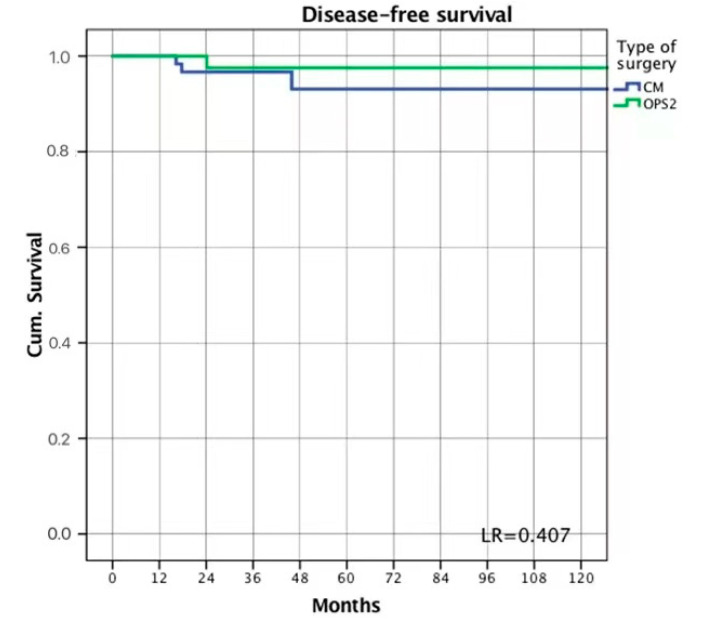
Disease-free survival.

**Table 1 cancers-14-05624-t001:** Type of Surgery and Breast Reconstruction.

Type of Surgery	N° (%)
**Mastectomy**	**91 (61.9%)**
Nipple-sparing mastectomy	75 (51%)
Skin-sparing mastectomy	14 (9.5%)
Skin-reducing mastectomy	2 (1.4%)
**Breast-conserving surgery**	**56 (38.1%)**
**Type of reconstruction**	**n° (%)**
**Mastectomy**	**91 (61.9%)**
Prepectoral implant	45 (30.6%)
Submuscular implant	28 (19%)
Tissue expander	17 (11.6%)
Tissue flap	1 (0.7%)
**Breast-conserving surgery**	**56 (38.1%)**
J mammoplasty	35 (23.8%)
T mammoplasty	9 (6.1%)
Round-block	9 (6.1%)
Racquet mammoplasty	2 (1.4%)
Batwing	1 (0.7%)

**Table 2 cancers-14-05624-t002:** Preoperative Tumor Characteristics.

Characteristics	Conservative Mastectomy	Level II Oncoplastic Surgery	*p* Value
**N° of patients**	91	56	
**Age (years)**	48.5 (47; 44–52)	50.2 (50.5; 45.3–54.8)	0.093
**Menopausal status (n°)**			0.176
Pre-menopausal	53 (58.2%)	27 (48.2%)	
Menopausal	29 (31.9%)	26 (46.4%)	
Perimenopausal	9 (9.9%)	3 (5.4%)	
**Familial Risk**			0.686
BRCA 1	2 (2.2%)	1 (1.8%)	
BRCA 2	3 (3.3%)	0	
Other high risk variant	6 (6.6%)	2 (3.6%)	
**BMI (kg/m^2^)**	22.3 (21.6; 20.1–23.9)	24.5 (23.5; 21.7–25.8)	**0.001**
**Bra size**			**<0.001**
1	6 (7.2%)	1 (1.9%)	
2	25 (30.1%)	2 (3.8%)	
3	40 (48.2%)	17 (32.1%)	
4	9 (10.8%)	21 (39.6%)	
≥5	3 (3.6%)	15 (11%)	
**Palpable lesion**			0.056
Yes	37 (41.1%)	15 (26.8%)	
No	53 (58.9%)	41 (73.2%)	
**Total lesion extension (mm) ^1^**	50.7 (50; 25–70)	46 (40; 30–55)	0.400
**Visible at ultrasound**			0.115
Yes	43 (47.3%)	20 (35.7%)	
No	38 (52.7%)	36 (64.3%)	
**Extension at ultrasound (mm)**	21.2 (20; 10–25)	16.1 (12.5; 10.3–20)	0.207
**Microcalcifications**			0.147
Yes	76 (83.5%)	51 (91.1%)	
No	15 (16.5%)	5 (8.9%)	
**Extension of microcalcifications (mm)**	48.5 (40; 20.3–67)	45.6 (40; 24–65)	0.953
**MRI appearance**			0.242
Negative MRI	11 (14.9%)	11 (28.9%))	
Mass lesion	8 (10.8%)	5 (13.2%)	
Non-mass lesion	49 (66.2%)	21 (55.3%)	
Both mass and non-mass lesions	6 (8.1%)	1 (2.6%)	
**Mass lesion extension (mm)**	20.3 (15.5; 10–25.8)	24.2 (22; 15–34.5)	0.257
**Non-mass lesion extension (mm)**	54.4 (60; 30–70)	44.7 (40; 30–55)	0.154
**Multifocality**			**0.005**
Yes	63 (69.2%)	26 (46.4%)	
No	28 (30.8%)	30 (53.6%)	
**Multicentricity**			**<0.001**
Yes	38 (41.8%)	2 (3.6%)	
No	53 (58.2%)	54 (96.4%)	
**Breast side**			0.210
Right	38 (41.8%)	28 (50%)	
Left	53 (58.2%)	28 (50%)	
**Breast quadrant**			0.818
Upper-outer	50 (54.9%)	29 (51.8%)	
Upper-inner	12 (13.2%)	10 (17.9%)	
Lower-outer	8 (8.8%)	7 (12.5%)	
Lower-inner	18 (19.8%)	8 (14.3%)	
Central	3 (3.3%)	2 (3.6%)	
**Type of biopsy**			**0.047**
Stereotactic	45 (51.1%)	37 (72.5%)	
Ultrasound	32 (36.4%)	14 (27.5%)	
Other (MRI, excisional)	11 (12.4%)	0	

^1^ Defined as widest diameter detected preoperatively in all imaging modalities.

**Table 3 cancers-14-05624-t003:** Final Histology Tumor Characteristics.

Characteristics	Conservative Mastectomy	Level II Oncoplastic Surgery	*p* Value
**Specimen volume (cm^3^)**	275.4 (221; 141.3–329.7)	137.8 (96.9; 52.3–224.5)	**<0.001**
**Tumor extension (mm)**	46.7 (40; 25–60)	38.1 (35; 25–45)	**0.037**
**Tumor Grade**			0.266
G1	11 (12.1%)	5 (8.9%)	
G2	33 (36.3%)	28 (50%)	
G3	47 (51.6%)	23 (41.1%)	
**Comedonecrosis**			1.000
Yes	59 (64.8%)	36 (64.3%)	
No	32 (35.2%)	20 (35.7%)	
**Multifocality**			0.861
Yes	56 (61.5%)	36 (64.3%)	
No	35 (38.5%)	20 (35.7%)	
**Multicentricity**			**0.001**
Yes	26 (28.6%)	3 (5.4%)	
No	65 (71.4%)	53 (94.6%)	
**Tumor histology subtype**			
Non special type	38 (41.8%)	28 (50%)	0.394
Cribriform	20 (22%)	11 (19.6%)	0.836
Papillary	25 (27.5%)	13 (23.6%)	0.699
Solid	29 (31.9%)	11 (19.6%)	0.128
Apocrine	9 (9.9%)	10 (17.9%)	0.207
Clinging	3 (3.3%)	2 (3.6%)	1.000
Lobular pleomorphic	1 (1.1%)	1 (1.8%)	1.000
Other	5 (5.5%)	5 (8.9%)	0.711
**Tumor biologic subtype**			
Hormone positivity	58 (63.7%)	44 (78.6%)	0.148
Triple negative	3 (3.3%)	1 (1.8%)	0.442
HER2 positive	36 (39.6%)	19 (33.9%)	0.789
**Microinvasive**	16 (17.6%)	12 (21.4%)	0.666
**Lymph node status**			**0.001**
pNx	6 (6.6%)	16 (28.6%)	
pN0	77 (84.6%)	38 (67.9%)	
Other (pN0i+, pN1mi, pN1)	8 (8.8%)	2 (3.6%)	

**Table 4 cancers-14-05624-t004:** Surgical and Oncological Outcomes.

Surgical Outcomes	Conservative Mastectomy	Level II Oncoplastic Surgery	*p* Value
**Margins**			0.211
Involved	8 (8.8%)	6 (10.7%)	
Close	18 (19.8%)	5 (8.9%)	
>2 mm	65 (71.4%)	45 (80.4%)	
**Perioperative complications**	16 (17.6%)	9 (16.1%)	0.827
**Clavien–Dindo classification**			0.809
Grade 1	12 (13.2%)	8 (14.3%)	
Grade 2	2 (2.2%)	0	
Grade 3	2 (2.2%)	1 (1.8%)	
**Hospital stay (days)**	3.51 (3; 3–4)	2.91 (3; 2–3)	**<0.001**
**Additional subsequent surgery**	30 (33%)	4 (7.1%)	**<0.001**
re-excision	1 (1.1%)	1 (1.8%)	1.000
local recurrence	3 (3.3%)	1 (1.8%)	0.662
perioperative complication	2 (2.2%)	1 (1.8%)	1.000
cosmetic adjustment	24 (26.4%)	1 (1.8%)	**<0.001**
**Oncological Outcomes**	**Conservative Mastectomy**	**Level II Oncoplastic Surgery**	***p* Value**
Local recurrence	3 (3.3%)	1 (1.8%)	0.662

**Table 5 cancers-14-05624-t005:** Patient-reported outcomes (data obtained from the 124 patients who completed all surveys).

**BREAST-Q©**				
**Module**	**N° of Questions**	**Conservative Mastectomy Score ***	**Level II Oncoplastic Surgery Score ***	** *p* ** **Value**
**Psychosocial well-being ^1^**	10	64.1 (62; 47–82.3)	71.3 (71; 51–100)	0.116
**Physical well-being ^2^**	10	31.6 (32; 20–45)	27.8 (26; 8–43)	0.305
**Sexual well-being ^1^**	6	48.2 (48; 41–59)	59 (60.5; 43–79)	**0.015**
**Satisfaction with breasts ^1^**	4	63.9 (64; 48–82)	71.7 (71; 53–100)	0.064
**Residual Skin Sensitivity ^2^**				
**Question**	**Score Range**	**Conservative Mastectomy Score**	**Level II Oncoplastic Surgery Scores**	** *p* ** **Value**
**How much skin sensitivity have you lost?**	1–10	6.7 (7; 5–9)	4 (3.5; 1–6)	**<0.001**
**How much does skin sensitivity loss influence your everyday life?**	1–10	3.8 (3; 2–6)	2.3 (1; 1–3)	**<0.001**
**How much does skin sensitivity loss influence your sexual life?**	1–10	5.3 (5.5; 2–8)	3.6 (3; 1–6)	**0.002**

* score is expressed in equivalent Rasch transformed score as per BREAST-Q© conversion tables, every score ranges from 0 to 100; ^1^ Higher scores reflect better outcomes; ^2^ Higher scores reflect worse outcomes.

## Data Availability

Data available on request due to privacy restrictions.

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
