# Peer review of "Oncoplastic Breast Surgery versus Conservative Mastectomy in the Management of Large Ductal Carcinoma In Situ (DCIS): Surgical, Oncological, and Patient-Reported Outcomes"

_cancers, 2022, doi:10.3390/cancers14225624_

Round 1
Reviewer 1 Report
I do congratulate the authors for this study which aims to compare level II oncoplastic surgery to mastectomy in the treatment of large DCIS. Surgical, oncological, and patient-reported outcomes in DCIS treated with OPS2 are analyzed. The idea is welcomed, as there are few if any studies published on this aspect. 943 patients were initially selected and after exclusion criteria filtering number 147 patients were included. The study design and research protocol are well-designed. results are clearly presented and discussions support the scientific soundness of the paper. The study conclusions suggest that breast-conserving surgery with OPS2 is safe and 346 feasible in selected patients with large DCIS, as it provides results comparable to CM in terms of oncological and surgical efficacy.
Author Response
We wish to thank the reviewer for his kind words on our manuscript.
Reviewer 2 Report
The manuscript submitted by Elena Jane et al. demonstrated level II oncoplastic surgery (OPS2) is more beneficial for patients with ductal carcinoma in situ (DCIS) than conservative mastectomy. They investigate clinical data, histology characteristics, and surgical and post-operate data in a 147 cohort. In addition, they involved patient-reported outcomes and skin sensitivity to consider physical and psychosocial impacts on life quality. Their data support their conclusion well. However, the publication's quality can be improved with the following clarifications.
- How to define large DCIS?
- DCIS is a non-invasive disease and theoretically does not spread to axillary lymph nodes. Other research found the incidence of sentinel lymph node metastasis in cases of pure DCIS was 0.39%. However, in table 1, the lymph node status seems too high.
- At Line 180, there is no Fig2 in this manuscript.
- OPS2 resections of large lesions involving up to 50% of the total breast volume. What percentage of these part of patients? CM was performed if the estimated excision volume was >50%. And the CM group has more multicentricity. Dose these patients have high risk? If a patient with multicentricity DCIS and the estimated excision volume was >50%, dose there any option for surgery? Does OPS2 have a benefit for this patient as well?
- What does perioperative impact include?
- They mention that nearly half of the patients who underwent CM or OPS2 for a preoperative diagnosis of extensive DCIS were found to have an invasive component at final histology. As this study is retrospective, they exclude this part of patients. But in clinical routine, it could be difficult to know if DCIS with an invasive component before surgery. How to make the decision to choose OPS2? What if an IDC was found in the final histology report, what is the following treatment?
Author Response
1. How to define large DCIS?
That is an excellent question! The answer is definitely more complicated than expected. In fact, in the end we chose not to specify a definition for large DCIS because in our retrospective study patients were selected based on surgery, not lesion extension. Data show that mean lesion extension was above 4cm, but, as specified in the Methods section, the feature guiding the surgical choice was not lesion extension but tumor to breast ratio. We still chose to include the term “large DCIS” in the paper title, as it is indicative of the disease feature in a quick way.
2. DCIS is a non-invasive disease and theoretically does not spread to axillary lymph nodes. Other research found the incidence of sentinel lymph node metastasis in cases of pure DCIS was 0.39%. However, in table 1, the lymph node status seems too high.
Thank you for your correct insight. We believe that nodal involvement in our population was unusually high because of the unique aggressiveness associated with extensive DCIS. Although an extensive explanation on nodal staging was beyond the purpose of this manuscript, we added a brief paragraph in the Discussion section addressing our unusually high rate of nodal involvement. We believe the manuscript has indeed been enhanced by this addition, so thank you for the insight!
3. At Line 180, there is no Fig2 in this manuscript.
We apologize for the inconvenience, the figure had been lost in the various editings. It has now been added.
4. OPS2 resections of large lesions involving up to 50% of the total breast volume. What percentage of these part of patients? CM was performed if the estimated excision volume was >50%. And the CM group has more multicentricity. Dose these patients have high risk? If a patient with multicentricity DCIS and the estimated excision volume was >50%, dose there any option for surgery? Does OPS2 have a benefit for this patient as well?
Again, a very valuable insight. We added a paragraph in paragraph 4.1 of the Discussion section addressing this topic.
5. What does perioperative impact include?
We clarified the meaning of “perioperative impact” in line 327.
6. They mention that nearly half of the patients who underwent CM or OPS2 for a preoperative diagnosis of extensive DCIS were found to have an invasive component at final histology. As this study is retrospective, they exclude this part of patients. But in clinical routine, it could be difficult to know if DCIS with an invasive component before surgery. How to make the decision to choose OPS2? What if an IDC was found in the final histology report, what is the following treatment?
IDC with concurrent DCIS represents a distinct disease, therefore we chose to exclude the patients with this final diagnosis in order to focus our study on the treatment of pure DCIS. While you make a very good point of an interesting topic open to a very rich discussion, we believe it to be beyond the purpose of this manuscript. We will consider continuing our study by including all 104 patients with invasive disease at final histology and assessing the outcomes of this population.
Reviewer 3 Report
This study aimed to evaluate differences in OPS2 and mastectomy for large volume DCIS. It included evaluation of margins, re-excision, re-operation, oncologic and patient reported outcomes. This is an important and well-done study that shows no differences in surgical/oncologic outcomes, but improved patient reported outcomes in the OPS2 group.
Table 2 - overall will defer to the journal for formatting but I find the format hard to read. Would prefer to see the first column left justified and aligned.
Table 2, Palpable Lesion, No - there is a , rather than . for 73,2%
Table 2, Total lesion extension - please be more specific here. Is this all imaging modalities?
Author Response
Thank you for your kind comments. We have fixed the issues as follows:
Table 2 - overall will defer to the journal for formatting but I find the format hard to read. Would prefer to see the first column left justified and aligned.
- We agree, we formatted the tables as you suggested.
Table 2, Palpable Lesion, No - there is a , rather than . for 73,2%
- Thank you for noticing, it has been corrected
Table 2, Total lesion extension - please be more specific here. Is this all imaging modalities?
- Yes. We realize it could have been unclear and added a note in the table footer